# A New Removable Helical Metallic Stent for the Treatment of Tracheomalacia in Children: Study in Pathological Animal Model

**DOI:** 10.3390/jcm11226757

**Published:** 2022-11-15

**Authors:** Sandra Lopez-Minguez, Sergio Rodriguez-Zapater, Cristina Bonastre, Jose Rodriguez, Miguel Angel De Gregorio, Jose Andres Guirola, Carolina Serrano-Casorran

**Affiliations:** 1Department of Animal Pathology, Faculty of Veterinary, University of Zaragoza, Miguel Servet 177, 50013 Zaragoza, Spain; 2Minimally Invasive Research Group (GITMI), University of Zaragoza, Miguel Servet 177, 50013 Zaragoza, Spain; 3Lozano Blesa University Hospital, 50009 Zaragoza, Spain

**Keywords:** animal testing, non-traumatic tracheal stent removal technique, removable stent, tracheomalacia, pediatrics

## Abstract

Background: Congenital tracheomalacia is a pathology with no consensus of medical or surgical approach. The permanent nature and the major complications associated with metallic stents have limited their use over the years. The purpose of this study was to evaluate the feasibility of a helical stent design removal. Methods: Ten dogs diagnosed with tracheal collapse and treated with the helical stent were involved in the study. Animals were classified into three groups depending on stent indwelling time. Prior to the removal, endoscopic evaluation was performed to assess endothelization grade, mucous accumulation, and the presence of stenosis. During the removal, bleeding, fracture, or impossibility of removal were noted. After the removal, all macroscopic mucosal changes were recorded. Results: Technical success was 100%, without any complications. Complete epithelization of the stent was visualized in 7/10 animals. The removal procedure duration ranged from 2–12 min. At post-removal endoscopy, bleeding or epithelial damage, was visualized in any case. Stent fracture during removal occurred in one animal. Conclusions: The removal of a metallic stent with spiral geometry is feasible, simple, and without complications, regardless of the degree of neo-epithelialization.

## 1. Introduction

Congenital tracheomalacia (TM) is a pathological condition characterized by a collapse of the airway due to a loss in the integrity of the tracheal cartilage. Although it is considered a rare disease, it is the most common congenital abnormality of the trachea with an incidence of at least 1 in 2100 children [1]. Depending on the side affected and the grade and dynamism of the stenosis, the symptoms may be presented with a wide spectrum from noisy breathing to potential acute life-threatening respiratory distress [2]. In the same way, the management varies from a conservative approach to complex interventions, depending on the extension, localization, and grade of the stenosis. There is no standard treatment for tracheomalacia, the majority of patients have self-limited disease. However, when necessary, the positive pressure ventilation of 5–10 cm H_2_O [3], which prevents collapse during expiration raising the intraluminal pressure, is a viable and safe option [4]. Endoscopic implantation of stents has been considered for the treatment of TM. The conceptual advantages of stents are based on their less invasive nature and easier surgical approach. Filler et al. in 1986 [5] first reported the endoscopic treatment of the airway with metallic stents. Since then, several professionals have described their experience with a high diversity of results.

There are numerous lines of research open to find a satisfactory alternative for neonates. Generally, research on the airway is performed in lab animals [6]. Nevertheless, in the case of tracheomalacia, there is a similar pathology present in veterinary patients, canine tracheal collapse (CTC), that can provide real conditions for tracheal stent studies [7]. In veterinary, the etiology is not well-understood, complex, and multifactorial [7,8,9,10]. As in humans, similar major complications with tracheal stents are present [11].

Metallic stents are considered permanent endoluminal prostheses due to their rapid epithelization and the difficulties, or impossibility, of removal after a short period of time [12,13]. Complications after its implantation include chronic tracheitis, tissue ingrowth, mucous accumulation, bacterial infection, stent fracture and, the erosion of the tracheal wall and creation of an arterial fistula [14]. Granulation tissue formation and increased secretions have been the most reported complications [12,13]. To avoid prostheses removal, biodegradable stents have been developed. Biodegradable designs tend to fragment during the degradation process and these fragments can block the distal airway. Recently, a novel metallic helical stent has been developed for the treatment of tracheomalacia, with great preliminary results obtained in laboratory animals and in canines with TM [15,16].

The purpose of the present study was to determine the feasibility of the metallic helical stent (MSS) removal at different indwelling times in dogs with TM. We hypothesized that the helical geometry would enable its removal without damaging the tissue, regardless of the time elapsed since implantation.

## 2. Materials and Methods

### 2.1. Case Selection and Study Design

The study presented is a retrospective non-randomized research performed in canines diagnosed of tracheal collapse. All the procedures were approved by the Animal Ethics Experimentation Committee of the University of Zaragoza (PI50/18) and elaborated according to the Animal Research: Reporting of In Vivo Experiments guidelines [17].

Inclusion criteria were dogs diagnosed with TM that underwent a new MSS implantation and the removal was performed at least 7 days after the procedure.

Registered patient data included weight, body condition score (BCS), tracheal collapse grade, and bronchial affection. Tracheal measurements were performed under fluoroscopy with the established parameters described in veterinary medicine [7,8].

Endoprostheses MSS (patent ES2725273) were manufactured in super-elastic single NiTi wires of 0.3 mm. The same shape and measurement of MSS were implanted in all animals; conic shape of 12 × 10 mm in diameter and 100 mm in length. The cranial tip of the stent had a small hole for fixation to the tracheal wall.

The dogs were classified into three groups depending on the indwelling time of MSS removal. Prior to 30 days, post-implantation was considered as short-term (<30 days), between 1 and 6 months was considered as intermediate (31–180 days), and more than 12 months post-MSS was considered as long-term (>365 days).

The implantation procedure was performed as previously described [18]. Post-surgical medical prescriptions included corticosteroids (reduction regimen from 1 mg/kg/12 h to 0.25 mg/kg/24 h) for 21 days, inhaled fluticasone propionate 50 µg every 12 h for 7 days, codeine (1.5 mg/kg/8 h/7 days), and an antibiotic (amoxicillin clavulanic 12.5 mg/kg/12 h) for 7 days.

### 2.2. Endoscopic Evaluation Prior to Removal

Prior to the removal procedure, an endoscopic examination was performed to evaluate the re-epithelization grade of the stent, tracheal congestion, mucous retention, the presence of granuloma, stenosis, and the areas without contact between the MSS and the tracheal wall.

Endoscopic tests were performed under general anaesthesia using as premedication a combination of acepromazine (0.04 mg/kg), butorphanol (0.3 mg/kg) IM, and as inductor alfaxalone (dose-effect) IV. During the test, oxygen was supplied through an intratracheal cannula.

An endoscopic optic of 30° and 4 mm in diameter (Karl Storz, Hopkins II) was used, connected to a Karl Storz-Endoscope Telecam SL pal 202120 20 camera, with a Karl Storz-Endoscope Xenon nova 300 201340 20 light source. An endoscopic optic of 30° and 4 mm in diameter (Karl Storz, Hopkins II) was used, connected to a Karl Storz-Endoscope Telecam SL pal 202120 20 camera.

Re-epithelization was graded into three groups: Naked MSS, <10% of epithelization (grade 0); partial epithelization, >10% to <80% of the MSS coated (grade 1); and total epithelization, >80% of the MSS epithelized (grade 2). Mucous retention was classified as 0, non-retention; 1, moderate, isolated liquid mucus; and 2, profuse amounts, ranging from a colorless to white appearance. Both of these characteristics were grouped according to previous classifications [6].

The contact between the MSS and the tracheal wall was divided into three groups: Both sides without contact (Group 0); one side without contact (Group 1); and the entire stent in contact (Group 2) (Figure 1). In the same way, the presence of granuloma was classified as 0, when no evidence was observed; 1, if only one granuloma was present; and 2, when more than one granuloma was present. Finally, tracheal congestion and stenosis were registered as present or absent.

### 2.3. Removal Procedure, Classification, and Complications

Stent removal was performed under endoscopic visualization with a minimal invasive aggression of the tissue. A forceps was used for assistance to stretch the cranial tip of the MSS through the working channel of the endoscope for the removal. Under endoscopic visualization, the unscrewing of the stent was observed, and the removal time was recorded.

Animals that had a fixation of the MSS, it was cut prior to the procedure. Moreover, when the MSS was completely removed, the presence or absence of a mucosal macroscopic finding (MSS imprint) on the tracheal wall was evaluated (Figure 2).

Removal decisions were categorized as fracture of the MSS (FR) (Figure 3), movement of the MSS (M), and inadequate stent length (SL).

Complications, such as bleeding during or after the extraction, breakage of the prosthesis when pulling, and the impossibility of extraction were noted.

### 2.4. Data Analysis

Baseline descriptive statistics were described as the mean and standard deviation (SD) for normally distributed variables, and non-normally distributed variables as the median and range. The normality was analyzed with the Shapiro–Wilk test. Between groups, frequencies were compared using the likelihood ratio test, and the quantitative variables were analyzed using *t*-Student test or Kruskal–Wallis test.

## 3. Results

A total of 10 dogs classified as “toy breed” underwent MSS removal. The removals were performed between the first 7 days and 30 months post-implantation; removal associated complications were not observed in any case.

The mean body weight was 4.61 kg (SD: 1.603 kg). Only 1/10 individual was classified as normal weight (BCS III), 5/10 were graded as BCS IV, and 4/10 were categorized as BCS V.

The most observed CTC grade was IV (60% [6/10]), followed by grade III (30% [3/10]); only one animal presented as grade II. First endoscopic evaluation demonstrated the absence of bronchial collapse (40% [4/10]), left bronchial collapse (30% [3/10]), and collapse in both bronchi (30% [3/10]), with a significant difference between groups (*p* = 0.049). The mean tracheal size in the cervical area was 10.7 mm (SD: 1.24 mm), the thoracic inlet was 8.8 mm (SD: 1.35 mm), and the intrathoracic region was 9.9 mm (SD: 1.29 mm). The average tracheal length was 93.86 mm (SD: 15.57 mm). All general data related to the characteristics of the patient are presented in Table 1.

### 3.1. Group 1 (n = 2)

MSS removal was required after a mean of 14.50 ± 19.09 days, with an average procedure duration of 3.50 min (SD: 2.12 min); the removal decision was classified as M (movement) in all cases. The main cause of this phenomena was an underestimated diameter of the stent. Endoscopic evaluation revealed no epithelization in 100% (2/2), MSS imprint in 50% (1/2), and tracheal congestion in both animals (100% [2/2]).

### 3.2. Group 2 (n = 5)

The mean removal indwelling time was after 151 days (SD: 54.07 days), with a mean procedure duration of 6.75 ± 3.25 min. Four animals required stent exchange by a longer one (4/5), and one individual had stent fracture. Stent fracture was at 210 days after implantation due to a defect in the material (damage caused by the insertion forceps).

Endoscopically, stent epithelialization was considered as grade 1 in 1/5 animals, and as grade 2 (totally epithelized) in 4/5 of the animals. The presence of a granuloma was observed in the animal with stent fracture. The MSS imprints were observed in all animals (100% [5/5]) of this group.

### 3.3. Group 3 (n = 3)

This group was represented by three animals, with the MSS removal date 661.66 days (SD: 289.23 days) post-stenting. Two dogs needed a MSS exchange due to pathology progression, and one had stent fracture. Full epithelialization was observed in all cases, without tissue alteration, and with a visible MSS imprint after the removal. The fracture of the MSS during the removal was observed in one animal.

The summary of removal decision and the characteristics observed are reflected in Table 2.

## 4. Discussion

TM is a benign airway structural anomaly that causes respiratory distress. This condition is caused due to a developmental defect in the cartilage of tracheal rings, that can be accompanied by an hyperlaxity of the pars muscular. Most children under this condition may not need surgical intervention, and the pathology may be solved spontaneously before 2 years of age [19,20]. The weak cartilage of the trachea tends to be more rigid when they growth, and symptoms subside [21,22]. For those children that require an intervention to palliate respiratory distress or recurrent infections, an exhaustive study of vascular abnormalities and malacic extensions should be performed. Currently, there is no consensus on medical or surgical approach [23].

The use of endotracheal stents immediately restores airway patency. There are numerous types and morphologies of tracheal stents; each one presents advantages, but all of them are accompanied by major or minor complications. In 2005, the Food and Drug Administration published an advisory on the use of metallic tracheal stents in patients with benign airway disease [24]. The complete re-epithelization of uncovered metallic stents, the tissue overgrowth, and/or granuloma formation, advise against its use. In adults, silicone stents are preferred. These prostheses are a rigid plastic tube that easily migrate and prevent mucociliary clearance. These phenomena increase recurrent infections and the mucous accumulation can obstruct the intern lumen of the tube [25,26,27]. The thickness of the tube exacerbates these complications, which absolutely prevents its use in neonates [12]. To find an alternative for children, uncovered and covered metallic stents have been used with similar results in adults. Uncovered stents were fully re-epithelized, maintaining ciliary clearance with a low-profile prosthesis. However, after a short period, the impossibility of removal without damaging the tissue precludes the continuous exchange as the child grows [13,14].

A study presented in 2016 [6] comparing nitinol, steel, and paclitaxel coated nitinol stents demonstrates that uncovered nitinol stents caused the minima complications. However, in no case it was possible to remove the stents 90 days after the implantation. In recent years, other authors [28] presented better results with covered stents. These stents maintain advantages, such as low profile, easier implantation, and removable. However, with silicone prostheses, there is an excessive mucous accumulation and tissue overgrowth over the end of the stent [29].

Recently, biodegradable stents are being studied. Polydioxanone stents are the most demanded due to their longest degradation [30]. A study of polydioxanone stents on rabbit’s trachea [31] demonstrated a total degradation after 90 days, and half degradation after 60 days. In no case the authors observed stent re-epithelization. This evidence agrees with the report by Sztanó et al. [32], in which the absence of re-epithelization during stent degradation induced the migration of some fragments to the lung with fatal results. Inflammatory tissue reaction and other epithelial abnormalities have been demonstrated to recover the normality once the stent has disappeared. However, these phenomena make the use of numerous stents necessary to maintain tracheal patency.

To the best of our knowledge, the present report is the first study performed in pathological canine that demonstrates long-term metallic stent removal [6,15,16,31]. The small size of canines and the high grade of tracheal collapse have allowed us to study the behavior of the MSS in conditions which are closer to pediatric patients who suffer from TM. Although CTC does not require stent removal, the permanent nature of metallic stents, and the complications associated over the years [33,34,35], have limited its use in dogs. Our results confirm the benefits of a spiral stent design. Correct measurements of the trachea, and selection of the appropriate stent dimensions, are fundamental and established parameters [11]. During this study, an MSS underestimation was the cause of the removal in Group 1. However, as observed in Group 2, when the difference between the trachea diameter and the MSS was not significant, it did not cause complications. The major complication observed in dogs after stent implantation was granuloma formation [36], and it was only visualized in one individual related to stent fracture. In contrast to what was expected in dogs, when the diameter of a stent is underestimated [37], our results did not show an association between the inappropriate size of the prosthesis and granulomas. Furthermore, the presence of gutters or mucous accumulation was not a frequent complication in our study, even though there were areas classified as having no MSS contact in three animals (3/10, 30%). Stents have a complex mesh-shape that seems to retain more mucous compared to a helical geometry [18]. Contrary to what was observed by other authors, the stent did not migrate in any case thanks to the fixation of the suture to the tracheal wall. Animals were classified as M of the stent, and the movement was always caudo-cranial. It was evidenced that when the stent was completely re-epithelialized, the extraction time was longer than naked MSS: 12 min compared to 2 min (SD: 5.17 min), respectively. We assume that the increase in removal time was related to a greater integration of the prosthesis in the tissue. Unscrewing in these patients offered more resistance, although it proceeded without complications.

Mondal et al. [15] recently published similar results about the use of a helical stent for tracheomalacia. In our experience, the helical geometry allows for a non-traumatic removal with an unscrewing rotatory movement. They evidenced a high tissue reaction in all animals, and fibrosis in some cases, which is far from our results. Another difference between the studies was the lab animal employed and the pathological condition of our animals, which provide an environment similar to children with tracheomalacia in contrast to the porcine healthy model [15].

Further studies are necessary with a larger sample of canine patients before human trial.

## 5. Conclusions

This study demonstrates the benefits of a spiral stent design in comparison with commercially available stents. MSS can solve some of the challenges of existing stents, thus it could be an alternative solution for the treatment of infant tracheomalacia. However, it must be taken into account that the trachea of an infant is smaller, both in diameter and length, than the trachea of canines. This implies that the observed complications during the present study could have different consequences in children that should be considered. Second, the screw-like geometry provides enough radial support to the airway while preserving mucociliary clearance, and the possibility of removal regardless of the re-epithelization.

## Figures and Tables

**Figure 1 jcm-11-06757-f001:**
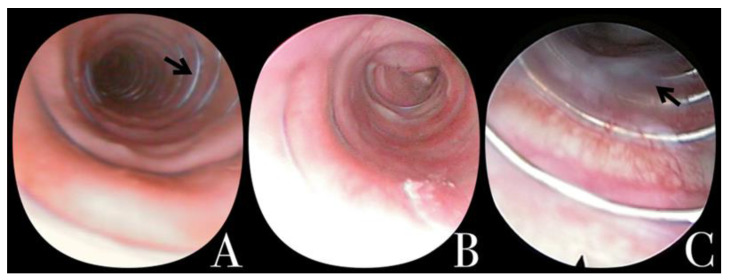
Tracheoscopy evaluation. (**A**) Naked MSS. No MSS contact (1) (arrow). (**B**) >80% MSS coated (2) (collapse of the canine can be observed caudal to the MSS). (**C**) Moderate mucous retention; isolated liquid mucous (1). MSS: Metallic spiral stent.

**Figure 2 jcm-11-06757-f002:**
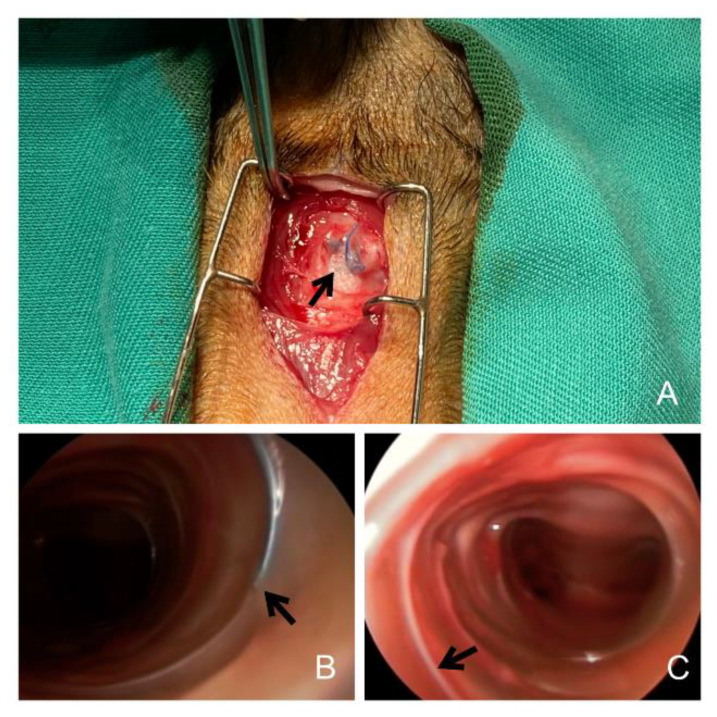
(**A**) Suture fixation of the MSS to the tracheal cartilage. (**B**) Unscrewing of the MSS. This visualizes how the prosthesis can be extracted without damaging the tissue. (**C**) White imprint after MSS removal. MSS: Metallic spiral stent.

**Figure 3 jcm-11-06757-f003:**
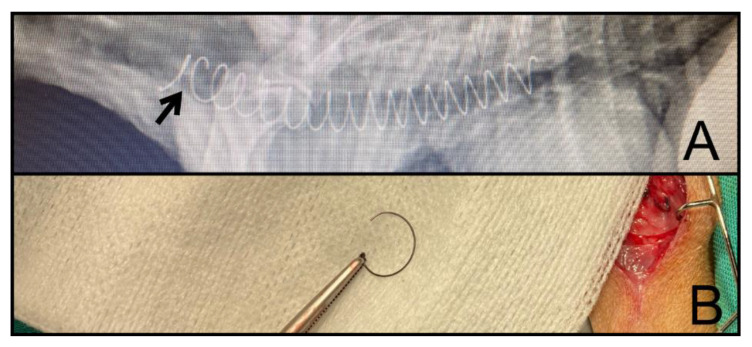
(**A**) Radiography of the fractured MSS. (**B**) Fragment removed after 220 days. MSS: Metallic spiral stent.

**Table 1 jcm-11-06757-t001:** Characteristics of the patients. General data recorded. Body condition score (BCS). Data are presented as the mean and standard deviation (SD) for normally distributed variables. Data are presented as the median and range for non-normally distributed variables.

		<30 Days	90–180 Days	>365 Days	Total	*p*-Value
*n*		2	5	3	10	
Body weight (kg)		6.23 (1.95)	4.37 ± 1.59	3.25	4.80 ± 1.67	0.213
BCS (I-V)	III	-	-	1/3 (33.3%)	1/10 (10%)	0.159
	IV	2/2 (100%)	3/5 (60%)	-	5/10 (50%)	
	V	-	2/5 (40%)	2/3 (66.6%)	4/10 (40%)	
TC grade	II	-	-	1/3 (33.3%)	1/10 (10%)	
	III	1/2 (50%)	2/3 (33.3%)		3/10 (30%)	
	IV	1/2 (50%)	3/5 (66.7%)	2/3	6/10 (60%)	
Bronchus collapse	No	2/2 (100%)	1/5 (20%)	1/3 (33.3%)	4/10 (40%)	0.053
	L	-	2/5 (40%)	1/3 (33.3%)	3/10 (30%)	
	L + R	-	2/5 (40%)	1/3 (33.3%)	3/10 (30%)	
Tracheal diameter (mm)	C	11.70 (2.00)	10.83 ± 0.42	8.5	10.73 ± 1.36	0.252
	I	10.25 (1.1)	8.13 ± 1.22	7.5	8.73 ± 1.47	0.172
	IT	10.60 (0.8)	8.87 ± 1.53	8.3	9.35 ± 1.41	0.172
Tracheal length (mm)		110.00 (20)	85.67 ± 5.13	75	92.00 ± 16.19	0.117

Bronchus collapse: No (absence), R (right main bronchus affected), L (left main bronchus affected). Tracheal diameter: C (cervical), I (Inlet), IT (intrathoracic).

**Table 2 jcm-11-06757-t002:** Mean results of endoscopic removal and its characteristics. Data are presented as the mean and standard deviation (SD) for normally distributed variables. Data are presented as the median and range for non-normally distributed variables.

		<30 Days	90–180 Days	>365 Days	Total	*p*-Value
Removal decision	FR	-	1/5 (20%)	1/3 (33.3%)	2/10 (20%)	0.081
	M	2/2 (100%)	-	-	2/10 (20%)	
	SL	-	4/5 (80%)	2/3 (66.6%)	6/10 (60%)	
Removal date	day	17.5 ± 14.84	160.00 ± 62.45	661.66 ± 289.23	279.72 ± 338.35	0.117
Removal Duration	min	3.50 ± 1.15	8.00 ± 4.00	10 ± 2.00	7.16 ± 3.32	0.213
Epithelialization	0 (naked)	2/2 (100.0%)	-	-	2/10 (20%)	0.081
	1 (partial)	-	1/5 (20%)	-	1/10 (10%)	
	2 (total)	-	4/5 (80%)	3/3 (100%)	7/10 (70%)	
SS imprint		1/2 (50.0%)	5/5 (100.0%)	3/3 (100%)	9/10 (90%)	0.268
Tracheal congestion		2/2 (100%)	1/5 (20%)	1/3 (33.3%)	4/10 (40%)	0.105
Secretion retention	1 (moderate)		1/5 (20%)		1/10 (10%)	
Granuloma	0 (absence)	2/2 (100%)	4/5 (80%)	3/3 (100%)	9/10 (90%)	0.452
	1 (unique)	-	1/5 (20%)	-	1/10 (10%)	
No MSS contact	0 (absence)	2/2 (100%)	1/5 (20%)	3/3 (100%)	7/10 (70%)	0.105
	2 (both sides)	-	4/5 (80%)	-	3/10 (30%)	
Stenosis		-	1/3 (33.3%)	-	1/10 (10%)	0.452

No significant differences have been observed. MSS: Metallic spiral stent. Removal decision: FR (MSS fracture), M (MSS movement), SL (exchange with a longer MSS). Removal dates are presented in days after the intervention date.

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
