# Peer review of "A New Removable Helical Metallic Stent for the Treatment of Tracheomalacia in Children: Study in Pathological Animal Model"

_jcm, 2022, doi:10.3390/jcm11226757_

Round 1

Reviewer 1 Report

In this report the authors describe the ability to remove a helical tracheal stent in dogs with tracheomalacia who had stent implantation for >7 days.  The authors divide the duration of stenting into short-term (<30 days), intermediate (31-180days), and long-term (>12 months).   The authors evaluate for complications/sequelae related to the stent including migration, stent fracture, epithelization, secretion retention, and granuloma.  Despite the presence of significant epithelialization in the majority of cases, the stents were able to be removed without significant  difficulty.  The ability to remove a stent that has been in place for a prolonged period is an important addition to the tracheomalacia literature as metallic stents notoriously provide challenges, and the authors highlight this nicely.   

While the primary goal of this study was to highlight the ability to remove the stents, the complications related to the stents seem to be down-played.  The authors mention that the use of a pathologic dog model provides an environment that is more similar to pediatric patients than the previous animal reports.  While this is likely true, complications such as stent migration and fracture are significantly more concerning in the pediatric airway which is approximately one-half to one third the length and diameter of the canine airway that are reported and thus likely more susceptible to respiratory compromise were these events to occur.  This should be expanded in the discussion. 

Page 1 line 32.  Larygnomalacia not tracheomalacia is the most common abnormality of the pediatric airway.  Perhaps amend to say the most “common congenital abnormality of the trachea” or something similar

Page 1 line 34-38, Tracheomalacia is a dynamic process not a fixed stenosis.  This should be adjusted accordingly as I don’t think the authors are evaluating tracheal stenosis which can also be managed with stenting to some degree.

Page 1 line 38, there is not standard treatment for tracheomalacian in children.  PEEP is a viable option for sure but is by no means the most common management strategy.  The majority of patient have self-limited disease and are not treated with any intervention.  

Perhaps the greatest concern of metallic stents is erosion and creation of an arterial fistula that can be fatal.  This should be included.  

There appear to be a few large reports of stent use in pediatric patients that are missing from the introduction that should be sited as they are quite important for the goals of this article eg Anton-Pacheco et al European Journal of Cardiothoracic Surg (2008) and Wang, Zhu, and Xu  Frontiers In Pediatrics (2020) among several others.  

The specific endoscopic equipment being used should be mentioned in the methods.  Adding information about anesthetic technique would be helpful as well.   Airway management during the procedure is also of critical interest.  

How was the airway measured?  I assume that this is by  radiography but am unable to tell. 

Table 2 is somewhat confusing to look at.  As a an example Epithelization is on two lines.  Also the use of 0,1, and 2 or 0,1  or 0,2 for categorical variables makes it difficult for the reader.  I’d suggest writing words rather than the numeric code in the table, though I do realize it’s in the text of the manuscript.  The 0,1, and 2 mean different things with each variable, so it makes the table unclear. Also SS imprint “10/6 (90%)” doesn’t make sense.   Perhaps that should be 9/10?

Author Response

Dear Reviewer,

Thank you very much for your comments, we hope that the contribution of removable stents will open up new therapeutic alternatives. We greatly appreciate your deep review of the article.

We have made the proposed revisions and we hope that the article meets your requirements.

In the attached file you will find the answers to the review point by point

Reviewer 2 Report

This study presents and gives an acceptable answer to a highly seen infantile/pediatric airway problem. İn our clinic, there are lots of infants with tracheomalacia and few of them need airway stents. We have no major complication with metalic airway stents which were inserted into trachea. Thoracic surgery unıt widely uses bronchial stents to adult lung transplant recipients for vanishing bronchus. Biodegradable stents are nice to use in many studies but can not be found in many countries. We need sometimes custom made ones for operated tracheoesophageal fistula with TM. This study nicely represents a minimally invasive way to treat TM in infants,too.

Author Response

Dear Reviewer,

Thank you very much for your comments, we hope that the contribution of removable stents will open up new therapeutic alternatives. We greatly appreciate your review of the article.

We have made some modifications to improve the report.

Regards